# High-throughput screening and validation of antibodies against synaptic proteins to explore opioid signaling dynamics

Mariana Lemos Duarte [1], Nikita A. Trimbake[1,2], Achla Gupta[1], Christine Tumanut[3], Xiaomin Fan[3], Catherine Woods[3], Akila Ram[4], Ivone Gomes[1], Erin N. Bobeck [4], Deborah Schechtman[5] & Lakshmi A. Devi [1✉]

Antibodies represent powerful tools to examine signal transduction pathways. Here, we present a strategy integrating multiple state-of-the-art methods to produce, validate, and utilize antibodies. Focusing on understudied synaptic proteins, we generated 137 recombinant antibodies. We used yeast display antibody libraries from the B cells of immunized rabbits, followed by FACS sorting under stringent conditions to identify high affinity antibodies. The antibodies were validated by high-throughput functional screening, and genome editing. Next, we explored the temporal dynamics of signaling in single cells. A subset of antibodies targeting opioid receptors were used to examine the effect of treatment with opiates that have played central roles in the worsening of the 'opioid epidemic.' We show that morphine and fentanyl exhibit differential temporal dynamics of receptor phosphorylation. In summary, high-throughput approaches can lead to the identification of antibody-based tools required for an in-depth understanding of the temporal dynamics of opioid signaling.

[1] Department of Pharmacological Sciences, Icahn School of Medicine at Mount Sinai, One Gustave L. Levy Place, Box 1603, New York City, NY 10029, USA. [2] Regeneron Pharmaceutical, 777 Old Saw Mill River Rd, Tarrytown, NY 10591, USA. [3] AvantGen Inc., 6162 Nancy Ridge Dr #150, San Diego, CA 92121, USA. [4] Department of Biology, Utah State University, Logan, UT 84322, USA. [5] Department of Biochemistry, University of São Paulo, 748 Av Prof Lineu Prestes, room 1208 Cidade Universitaria, São Paulo, SP 05508000, Brazil. ✉email: lakshmi.devi@mssm.edu

The majority of techniques used to investigate cell signaling rely at least in part on interactions with antibodies; however, we currently lack high-quality commercial antibodies against synaptic proteins and their post-translational modifications (Fig. 1). Given that immunoassays are essential tools for scientific investigation, this lack of highly specific antibodies drives the need to optimize their development[1].

The primary motivation to produce antibodies as reliable tools to explore differential signaling of synaptic proteins is related to the worsening of the 'Opioid Epidemic'. In the United States in 2017, 67.8% of fatal drug overdoses involved opioid usage[2]. Combating the opioid crisis will require a deep and detailed understanding of the molecular mechanisms underlying synaptic signaling in general, and opioid receptor signaling in particular. Here, we present a time- and cost-efficient strategy to produce, validate, and utilize rabbit recombinant antibodies (rAbs) to show dynamic modulation of ligand-directed signaling using two drugs of abuse, morphine, and fentanyl.

## Results and discusssion

**Recombinant antibody development and validation.** Our goal was to generate and characterize antibody clones for panels of antigens that were of interest due to their potential role in synaptic signaling (Fig. 1). As a strategy, we used a rational design for the selection of antigenic prototypic peptides that enhances immunogenicity[3], immunized rabbits for rAb generation since this allows for the production of rAbs with 10- to 100-fold higher affinity for antigen as compared to rAbs produced in mice[4]; and screened for high affinity antibodies using a yeast display system that allows for efficient FACS selection of antibody clones with high affinity and specificity[5,6] (Fig. 2a). This effort resulted in the generation of 137 rAbs recognizing 92 different synaptic proteins that exhibited high affinity in the nanomolar range ($K_d < 1$ nM; Fig. 2b). Some of these antibodies target phosphospecific residues in different proteins (Supplementary Data 3).

To validate these antibodies, we followed the criteria recommended by the International Working Group for Antibody Validation (IWGAV)[1]: (i) use of an orthogonal analysis to

validate antibody specificity in a number of cell types and tissues; (ii) use of proteins of interest expressed with an affinity tag to validate with known, well-characterized antibodies; (iii) use of genetic strategies to ensure that the antibodies do not yield a signal in knockouts of the target protein; and (iv) generation of antibodies to different epitopes in the same protein to assess antibody specificity.

The orthogonal analysis for rAb validation was carried out using two cell lines from different origins, namely Neuro2A (mouse neuroblastoma cells) and HEK293 (human embryonic kidney cells). The analysis of protein levels related to synaptic processes by ELISA indicated higher expression in Neuro2A cells in comparison to HEK293 (Fig. 2d). The ELISA data were compared to the reported relative mRNA expression in the NCBI database, and in GEO Profile (GDS5140 and GDS4233[7,8]). The correlation analysis between HEK293 and Neuro2A confirms the differential expression profile by mRNA and protein levels of these genes (Fig. 2c and Supplementary Table 1).

Next, we tested the suitability of the rAbs to study proteins in native tissues. For this, a western blot (WB) analysis was performed using protein extracts from total mouse brain and synaptosomal fractions. First, we investigated how the conditions for protein extraction affected the robustness of the rAb signal. WB was carried out with protein extracted under denaturing (Fig. 2e and Supplementary Fig. 1) and non-denaturing conditions (Fig. 2e and Supplementary Fig. 2). Using total mouse brain fractions, under denaturing conditions, we saw an increase in signal that was directly proportional to the protein concentration for 61 rAbs. Under non-denaturing conditions, 28 rAbs gave a robust signal. Of the nine antibodies tested using the synaptosome fraction, three gave a good, clean signal (Fig. 2e; Supplementary Figs. 1 and 2; Supplementary Data 3). These analyses revealed that the antibodies can be divided into three groups; (i) high specificity, where the blots contained a single band with the predicted molecular weight; 37 rAbs fell into this category, (ii) intermediate specificity, where the blots contained one high-intensity band corresponding to the predicted molecular weight but also contained additional bands; 27 rAbs fell into this category, and (iii) low specificity, when blots contained single or

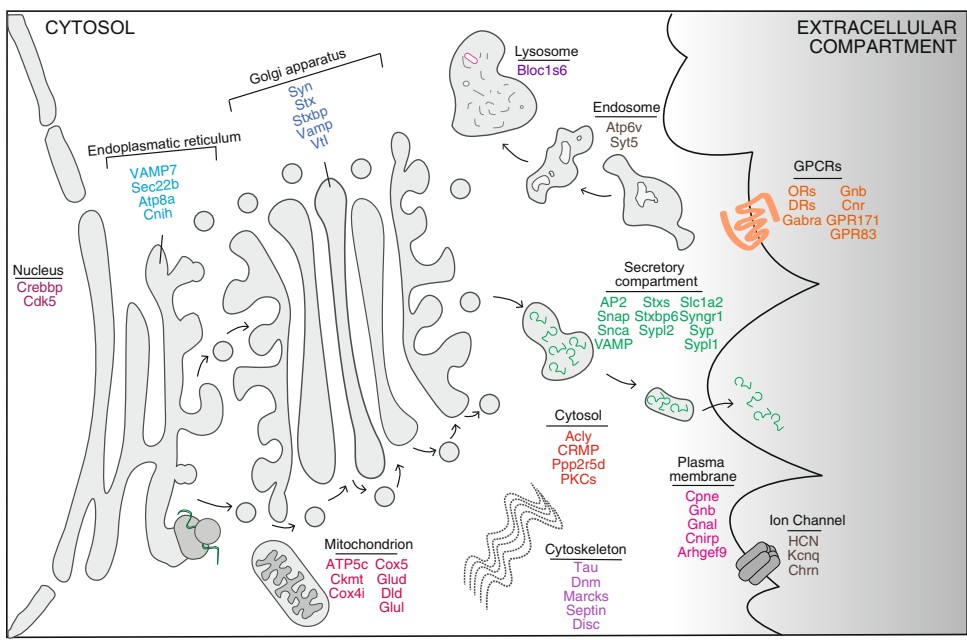

**Fig. 1 Schematic representation of synaptic related proteins used to generate antigenic peptides for production of recombinant antibodies.** The proteins and their subcellular localization are represented in the carton that highlights synaptic processes.

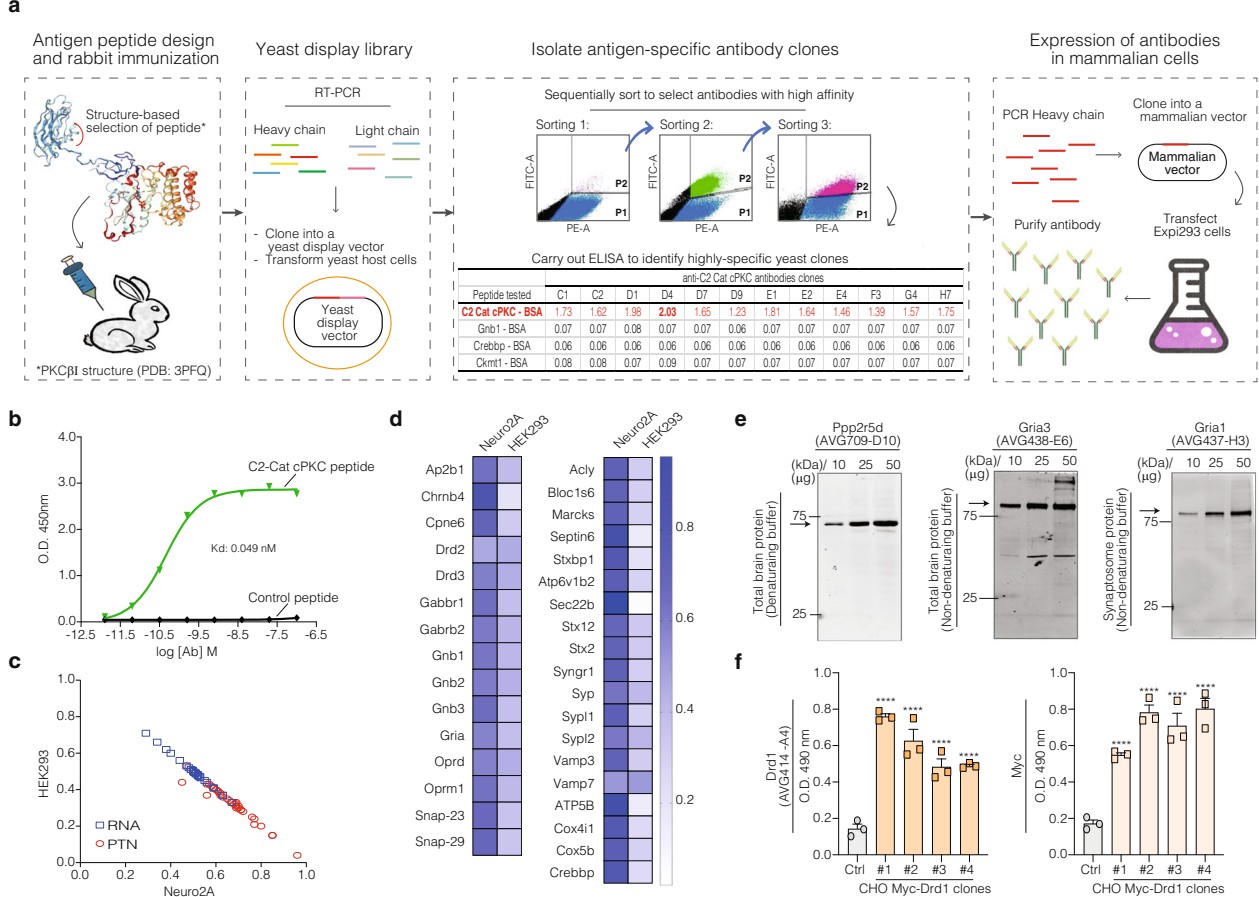

**Fig. 2 Workflow of recombinant antibody development and different approaches for its validation. a** Antibody development was stratified into four steps: peptide design[23] and rabbit immunization; construction of the yeast antibody display library; isolation of antigen-specific antibody clones and expression of recombinant antibodies in mammalian cells. **b** Binding curve of a purified antibody clone, anti-C2-Cat-cPKC (AVG459-D4), to immobilized C2-Cat-cPKC peptide compared to control peptide. **c** Correlation analysis between the relative expression levels of mRNA expression (blue square) and protein (red circles) in Neuro2A cells (x-axis) and HEK293 cells (y-axis). mRNA expression was obtained from the GeoProfile Dataset (GDS5140 for Neuro2A cells, and GSD4233 for HEK293 cells). The protein levels (PTN) were measured by ELISA using the newly developed antibodies. The relative expression is the mean of the sample measurement divided by the sum of the Neuro2A and HEK293 cell values (RANK for mRNA, O.D. for protein), from three experiments. $R^2$ 0.95 and p value < 0.0001. **d** Heatmap of relative protein expression from Neuro2A and HEK293 cells. Darker blue squares indicate higher expression levels from ELISA experiments. The ELISA was performed using the rAbs against the proteins listed in the heatmap. Data are mean ± SE of three experiments. **e** Western Blot analysis from total brain and synaptosomal protein extracts at different protein concentrations and buffers. **f** ELISA using CHO cells and different clones of CHO cells overexpressing Drd1 with a N-terminal Myc-tag. The ELISA was performed using antibodies specific for the Drd1 receptor (AVG414-A4) and for the Myc-tag. Data are mean ± SE of three experiments, P value < 0.0001; One-way ANOVA.

multiple bands with low signal; 25 rAbs fell into this category (Fig. 2e; Supplementary Figs. 1 and 2 and Supplementary Data 3). The biochemical properties of the proteins and their solubility under the extraction buffer, the presence of multiple protein isoforms, and the reduced protein expression in the sample tested have to be taken into consideration during the testing/validation of antibodies by WB.

Next, we used the epitope tag approach to validate the proteins of interest (members of the dopamine receptor family) by N-terminally tagging them with Myc, Flag, or HA tags. Cells overexpressing these tagged proteins were probed with our rAbs and cross-validated using commercially available antibodies to the epitope tag. The results show a great correlation between the signals from these two sets of antibodies (Fig. 2f and Supplementary Fig. 3).

**High-throughput microscopy to validate rAbs**. We also investigated the efficiency of the rAbs to stain single cells using immunocytochemistry (ICC). We tested two different ICC protocols: (i) 4% paraformaldehyde (PFA) to fix the cells followed by permeabilization with Triton-X-100, and (ii) 4% PFA followed by 100% methanol to fix the cells and permeabilization using saponin (Supplementary Fig. 4). Treatment with Triton-X-100, although widely used in subcellular characterization studies, is known to disrupt the structure of the plasma membrane leading to a loss in the resolution of membrane proteins. In contrast, saponin solubilizes cholesterol leading to a better resolution of membrane protein and vesicle compartments[9]. Since the majority of the rAbs tested were against membrane-associated synaptic related proteins (Fig. 1), the validation of the rAbs by immunocytochemistry was performed with the PFA/methanol and saponin protocol (Supplementary Figs. 5 and 6). These results shed light on the importance of exploring different immunostaining protocols to characterize the antibodies and the subcellular protein localization.

Previous studies have shown that immunostaining is a technique that works for many mAbs; however, immunostaining

is also notoriously prone to human bias[10]. To eliminate the concerns of bias, we developed a platform involving high-throughput microscopy (HTM) and machine learning that leads to an accurate, reproducible, unbiased, time- and cost-effective method of antibody validation (Supplementary Fig. 7). To perform the HTM, cells of interest were seeded on a 96-well plate; after immunostaining, images were acquired using the IN Cell microscope (GE Healthcare) and analyzed using CellProfiler 3.1.8 software[11,12]. To ensure that this method recognizes changes in cellular staining, we taught the software to initially identify the nuclei in a given image, and then to measure only the fluorescence in the cellular area surrounding the nuclei (thus eliminating erroneous measurements of background signal). To confirm the accuracy of this approach, we directly compared the results of our HTM/Cell Profiler platform with the commonly used ImageJ software analysis ($R^2$: 0.81, $p < 0.0001$; Supplementary Fig. 7). This platform avoids bias, increases time- and cost-efficiency, and allows for single-cell analysis.

Using this HTM/machine learning method, we determined the ability of the rAbs to detect the corresponding proteins in Neuro2A cells. We used the following criteria for the selection of rAbs (Fig. 3a), high fluorescence intensity or the fluorescence intensity should correlate with the dilution of the antibody[13] (Fig. 3b, c and Supplementary Data 4), and subcellular protein localization that corresponds to that reported in the published literature (Supplementary Figs. 5 and 6). Of the 133 rAbs tested, 91 fit the specificity criteria (Supplementary Data 3 and 4).

**Genome editing strategy to validate rAbs**. An important criterion to validate Ab specificity is to probe cells/tissues lacking the antigenic epitope[1,10]. We used the genetic strategy, CRISPR/Cas9 system to generate knockout (KO) Neuro2A cell lines for genes to three different signaling proteins: Protein Kinase C$\alpha$ (PKC$\alpha$), PKC$\beta$ and $\mu$-Opioid Recector (MOR) (Supplementary Fig. 8). These cells were then probed with rAbs; a lack of antibody signal was confirmed by immunostaining (Figs. 3–5 and Supplementary Fig. 9).

Since we had a number of rAbs to PKC$\beta$, the HTM/machine learning method was applied to confirm their specificity (Figs. 3c, d and 4b, and Supplementary Fig. 9a). A decrease in fluorescence intensity in the Neuro2A PKC$\beta$-KO cells compared to Neuro2A wild-type cells was observed in the case of 4 out of the 6 rAbs to PKC$\beta$ (Figs. 3d and 4b, and Supplementary Fig. 9a). We also assessed the specificity of the phosphospecific antibody clones to substrates known to be phosphorylated by PKC$\alpha$, namely MARCKS, Ppp2r5d, and MOR[14–17]. Neuro2A wild-type and Neuro2A PKC$\alpha$-KO cells (Supplementary Fig. 9b) were treated with PMA (phorbol 12-myristate 13-acetate), a known activator of PKC[18]. HTM analysis with the phosphospecific antibody clones revealed an increase in fluorescence only in the Neuro2A wild-type cells (Fig. 3e). We then repeated this experiment using another antibody clone directed against S82 of Ppp2r5d. This residue does not fit the PKC$\alpha$ phosphorylation consensus motif, and it has been described as a Janus kinase 3 (JAK3) and AKT substrate[19–21]. As expected, PKC activation via PMA did not lead to changes in signal by this rAb in Neuro2A PKC$\alpha$-KO cells (Supplementary Fig. 10). Together these results suggest that our integrative strategy for antibody development leads to high-specificity antibody clones, which could be successfully used to validate using a combination of genome editing, and HTM/machine learning approaches.

**Design and validation of conformation-specific antibodies for classical Protein Kinase C enzymes**. Next, we turned our attention to characterize the conformation-specific recombinant antibodies that preferentially recognize the active state of classical PKC (cPKC). This is especially important since there are no commercially available mAbs that can directly evaluate the active-state of cPKC. Kinases are highly dynamic, and structural changes are known to be related to different enzymatic states[22,23]. Inactive cPKC is found in a closed conformation, with the catalytic domain hidden by intramolecular interactions[22]. Active state cPKC has an open conformation with an exposed catalytic site, which includes the C2-domain[22] (Fig. 4a). Considering the conformational changes of cPKCs and their correlation with the activation state of these enzymes, previous studies described the successfull development of antibodies against the C2-domain of cPKCs and its specificity to recognize the active state of these kinases[24,25].

Here we report the development of two conformation-specific antibodies, one rAb against the C2-domain of PKC$\beta$, named C2-Cat-PKC$\beta$, and another against the antigenic sequence published in previous studies[24,25], that recognizes the C2-domain of all cPKCs in a similar fashion, named C2-Cat-cPKC (Fig. 4a). To validate these rAbs, we used HTM analysis to measure changes in fluorescence intensity in Neuro2A wild-type and PKC$\alpha$-KO or PKC$\beta$-KO cells. Neuro2A PKC$\beta$-KO cells showed a reduction in fluorescence intensity when probed with C2-Cat-PKC$\beta$ antibody (Fig. 4b).

Interestingly, Neuro2A wild-type cells treated with PMA showed an increase in fluorescence when probed with C2-Cat-cPKC rAb, and this was not seen in Neuro2A PKC$\alpha$-KO cells (Fig. 4c). As a control, we used a commercial antibody that does not distinguish between active and inactive PKC$\alpha$ and found no differences in fluorescence between control cells and cells treated with PMA (Fig. 4d). To confirm that the ability of the anti-C2-Cat-cPKC antibody to recognize active PKC is not cell-type specific and is PKC-selective, we used another neuroblastoma cell line (SK-N-SH) in the presence or absence of a known PKC inhibitor, Gö6983[26] (Fig. 4e). In these human cells, there was an increase in fluorescence upon PMA treatment as detected by the C2-Cat-cPKC antibody and this signal was completed eliminated by the inhibitor (Fig. 4e). Together these results support the idea that conformation-specific antibodies are invaluable tools to explore signaling dynamics[24,25,27].

**Validation of phosphospecific antibody for $\mu$-Opioid Receptor**. Next, we turned our attention to opioid receptor antibodies and evaluated the specificity of phosphospecific MOR antibodies using four complementary approaches: (i) use of CRISPR/Cas9 to knock out MOR in Neuro2A cells (Fig. 5a and Supplementary Figs. 8 and 9); (ii) use of phospho-site specific MOR mutant cells (Fig. 5b); (iii) use of MOR KO mouse tissue (Fig. 5c); and (iv) use of in vitro PKC$\alpha$-mediated phosphorylation of purified MOR (Fig. 5d).

Using HTM analysis, of the six antibodies tested, five showed a reduction in fluorescence staining in MOR KO cells (Fig. 5a and Supplementary Fig. 9). In addition, the specificity of the phosphospecific C-terminal MOR residues was investigated using phospho-site specific MOR mutants cells, S363A and T370A. MOR was N-terminally tagged with a pH-sensitive GFP (SpH)[28,29]. Neuro2A wild-type, Neuro2A S363A MOR, and Neuro2A T370A MOR cells were stained with anti-phosphospecific C-terminus MOR antibodies. To verify that the cells were expressing the MOR constructs, we also stained the cells with anti-GFP antibody (Fig. 5b). HTM analysis demonstrated a decrease in the fluorescence staining only in the mutant cells stained with the C-terminus MOR rAbs. As a complementary strategy we used membrane protein extracts from spinal cord tissue of MOR KO mice, and found a substantial reduction of signal in the WB probed with anti-phospho S375/T376 MOR antibody (Fig. 5c).

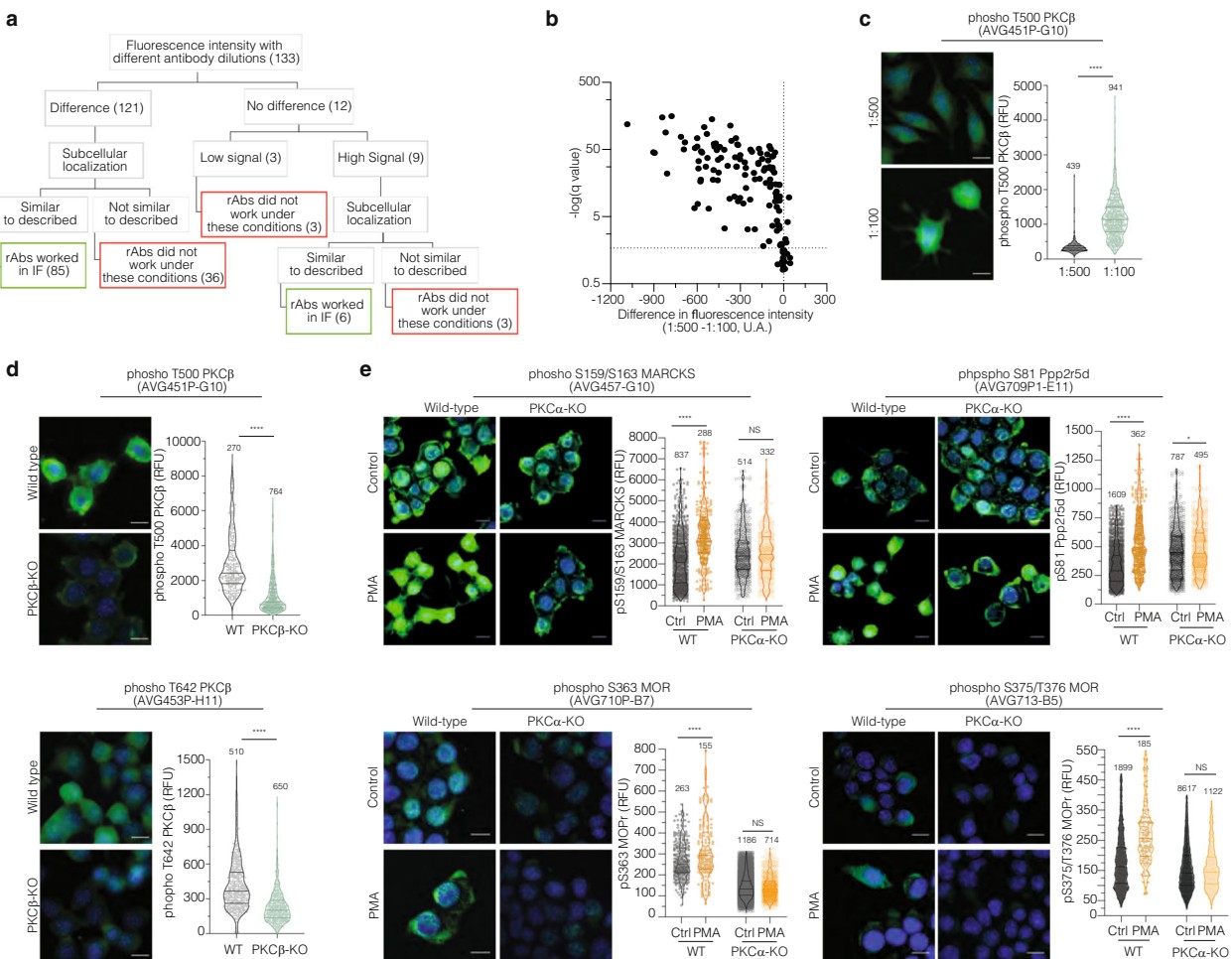

**Fig. 3 High-throughput microscopy and genome editing as strategies to validate rAbs. a** Schematic representation of the criteria used to validate the antibodies by high-throughput microscopy. The number of antibodies that matches each criteria step is represented in the box. **b** Volcano plot representing the fluorescence intensity difference obtained from the comparison of two rAbs dilutions. Each dot represents one antibody tested. The x-axis describes the difference in the antibody fluorescence intensity at two different dilutions (1:500–1:100), and the y-axis describes the log of the desired False Discovery Rate (Q). Using GraphPad prism, the two-stage step-up method of Benjamini et al.[13] was used to perform the analysis. The antibodies with Q higher than 2% were considered significant. **c** Representative image of Neuro2A cells probed with anti-phospho T500 PKCβ (AVG451P-G10) antibody at a dilution of 1:500 and 1:100 (green), and DAPI (blue). Images were acquired using the InCell microscope (×40) and fluorescence intensity measured by Cell Profiler software. Representative images with the antibody staining in green and DAPI in blue are shown. Volcano plot, each dot represents one cell. Lines are mean ± SD of two experiments, number of counted cells are indicated. RFU indicates the integrated fluorescence intensity units. ****p value < 0.0001; t-test. **d** Neuro2A and Neuro2A PKCβ-KO cells were probed with anti-phospho T500 PKCβ (AVG451P-G10) and anti-phospho T642 PKCβ (AVG453P-H11) antibodies (green) and DAPI (blue). Images were acquired and analyzed as previously described. ****p value < 0.0001; t test. **e** Validation of phosphospecific antibodies against PKC substrates. Neuro2A and Neuro2A PKCα-KO cells were probed with anti-phospho S159/S163 MARCKS (AVG 457-G10); anti-phospho pS81 Ppp2r5d (AVG709P1-E11) antibodies after PMA treatment (100 nM) for 30 min; anti-phospho S363 MOR (AVG710P-B7) antibody after PMA treatment (100 nM) for 60 min; anti-phospho S375/T376 MOR (AVG713-B5) antibody after PMA treatment (100 nM) for 30 s. Images were acquired and analyzed as previously described. ****p value < 0.0001; t test.

Finally, we carried out in vitro phosphorylation of purified MOR with recombinant PKCα. The purified MOR was assembled into nanodiscs, which mimic the membrane environment[30]. In this case, the nanodiscs containing MOR were assembled using brain lipid extract, 1-palmitoyl-2-oleoyl-sn-glycero-3-phosphocholine (POPC) and 1-palmitoyl-2-oleoyl-sn-glycero-3-[phospho-rac-(1-glycerol)] (POPG) as bilayer phospholipids, and MSP1D1, as a scaffold protein[30,31]. A WB analysis with anti-phospho S363 MOR and anti-phospho T370 MOR antibodies showed an increase in the band intensity in nanodiscs treated with recombinant PKCα (Fig. 5d). Although there are two bands, the high-molecular weight band (~75 kDa), as seen in previous reports[32,33], shows an increase in intensity when probed with these antibodies (following incubation with PKCα). These results

are consistent with published reports that suggest S363 MOR and T370 MOR as direct PKCα substrates[28,29,34–37].

**Temporal dynamics of cPKC activation by morphine and fentanyl**. The recent 'opioid epidemic' has refocused the interest of the field on understanding signaling by two important drugs of abuse, morphine and fentanyl[2,38]. Despite intense research efforts, the molecular mechanisms underlying the differential activation of opioid receptors by morphine and fentanyl are poorly understood[34]. Furthermore, PKCα has been shown to be important in opioid receptor desensitization[17,39,40] and opiate tolerance[17,41,42], but its role is not completely understood[34]. Using our antibody clones and the HTM/machine learning

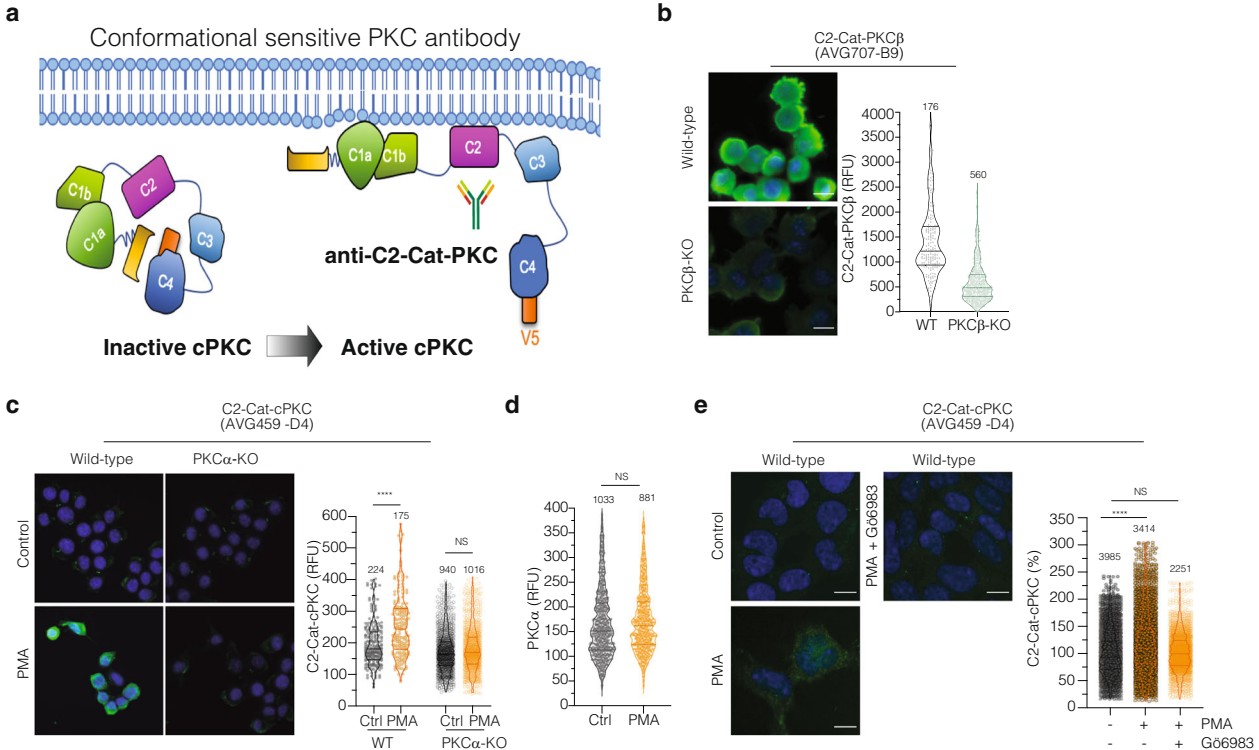

**Fig. 4 Validation of conformation-sensitive antibodies to classical PKCs. a** Schematic of conformational changes involved in generation of the active state of cPKC. Purple represents the C2 domain that becomes exposed after cPKC activation. This domain is recognized by two C2-Cat-cPKC antibodies: the C2-Cat-PKCβ antibody (specific for PKCβ), and the C2-Cat-cPKC antibody (recognizes all the classical cPKC isoenzymes). **b** Neuro2A and Neuro2A PKCβ-KO cells were probed with anti-C2-Cat-PKCβ (AVG707-B9) antibody. Images were acquired using the InCell microscope (×40) and fluorescence intensity measured by Cell Profiler software. Representative images with the antibody staining in green and DAPI in blue are shown. Volcano plot, each dot represents one cell. Lines are mean ± SD of two experiments, number of counted cells are indicated. RFU indicates the integrated fluorescence intensity units. ****$p$ value < 0.0001; $t$-test. **c** Neuro2A and Neuro2A PKCα-KO cells were treated with PMA (100 nM) for 30 min and probed with anti-C2-Cat-cPKC antibody. Images were acquired and analyzed as previously described. ****$p$ value < 0.0001; $t$-test. **d** Neuro2A cells were treated with PMA (100 nM) for 30 min and stained with anti-PKCα antibody (Santa Cruz; sc-8393). Images were acquired and analyzed as previously described. NS non-significant; $p$ value > 0.99; $t$ test. **e** SK-N-SH cells were treated with PMA (100 nM) with or without Gö6983 (1 μM) for 30 min and stained with anti-C2-Cat-cPKC antibody. Images were acquired and analyzed as previously described. ****$p$ value < 0.0001; $t$ test.

pipeline, we focused our next set of studies on elucidating the role that cPKC may play in differential opioid activation by morphine and fentanyl.

First, we confirmed the temporal dynamics of cPKC activation in SK-N-SH and Neuro2A cells. In these cells, treatment with PMA leads to a progressive increase in fluorescence starting at 1 min (Fig. 6a, d). These effects were abrogated by treatment with a PKC inhibitor, Gö6983, and in Neuro2A PKCα-KO cells (Fig. 6a, d, respectively).

Next, we examined the temporal dynamics of cPKC activation following treatment with morphine and fentanyl. Morphine treatment in Neuro2A or SK-N-SH cells leads to cPKC activation at 1 and 3 min (Fig. 6b, e) which returns to basal levels by 60 min (Fig. 6b, e, g). These effects were completely abrogated in Neuro2A PKCα-KO cells (Fig. 6e, g).

Interestingly, treatment with fentanyl in Neuro2A or SK-N-SH cells leads to an increase in PKC activation at 1 and 3 min (Fig. 6c, f), that is sustained until 60 min (Fig. 6c, f, g), and does not return to basal levels even after 120 min (Fig. 6h), suggesting important differences between the signaling responses to morphine and fentanyl. Furthermore, fentanyl-mediated cPKC activation is partially abrogated in the Neuro2A PKCα-KO cells (Fig. 6f, g, h), indicating that, in contrast to morphine, fentanyl may activate other classical PKC isoenzymes in addition to PKCα. Taken together, our studies highlight the differences in the temporal regulation of PKCα activity by morphine and fentanyl.

To assess whether the anti-C2-Cat-cPKC antibody could be useful to probe changes in signaling by endogenous opioid receptors in native tissue, wild-type mice were administered morphine and the extent of cPKC activation was evaluated. Acute administration of morphine (10 mg/kg, for 30 min) leads to activation of cPKC in the ventrolateral periaqueductal gray region and the magnitude of this increase is reduced upon chronic administration of morphine (10 mg/kg, twice daily for 4 days; Fig. 6i, j). These data are consistent with previous findings about the temporal dynamics of morphine-mediated PKC activation[24].

**Temporal dynamics of MOR phosphorylation by PKCα induced by morphine and fentanyl.** An important question in the field has been whether PKC activated by morphine versus fentanyl differentially phosphorylates MOR[36,43]. Using phosphospecific antibody clones against five different MOR residues: S363; T370, S375, T376, and T379 (Fig. 7), we examined the phosphorylation of these residues in Neuro2A wild-type and Neuro2A PKCα-KO cells. We find a decrease of basal phosphorylation of S363, T370, S375/T376 but not in T379 in Neuro2A PKCα-KO cells (Fig. 7a, d, g, h, i). This suggests that PKCα mediates the basal phosphorylation of S363, T370, S375, and T376 MOR residues but not of T379 MOR[34].

Next, we examined the time course of phosphorylation of different residues in MOR by global cPKC activity following treatment with PMA (Fig. 7a, d). The residue S363 in MOR has

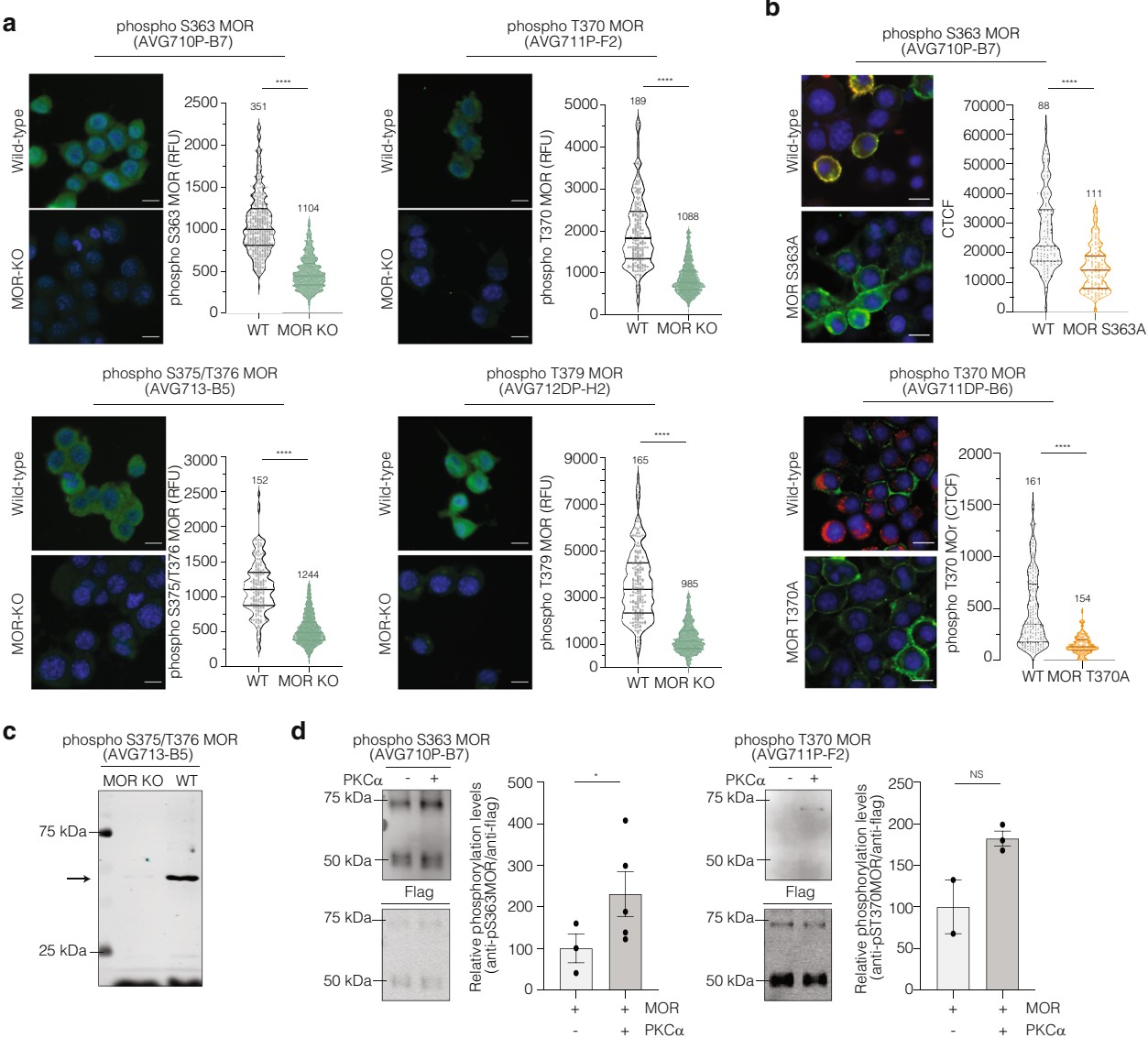

**Fig. 5 Validation of phosphospecific antibodies for MOR. a** Neuro2A and Neuro2A MOR-KO cells were stained with phosphospecific C-terminus MOR antibodies (green) and DAPI (blue). Cells were fixed with 4% PFA and permeabilized with 0.05% Triton-X-100. Representative images with the antibody staining in green and DAPI in blue are shown. Volcano plot, each dot represents one cell. Lines are mean ± SD of two experiments, number of counted cells are indicated. RFU indicates the integrated fluorescence intensity units. **b** Neuro2A, Neuro2A S363A MOR, and Neuro2A T370A MOR cells were stained with anti-phosphospecific C-terminus MOR (red) and anti-GFP (green) antibodies along with DAPI (blue). Cells were fixed with 4% PFA and 100% methanol, and permeabilized with 0.1% Saponin. Representative images are the merge of these three stainings. Images were acquired using the InCell microscope (×40) and fluorescence intensity measured using the Image J. Volcano plot, each dot represents one cell. Lines are mean ± SD of two experiments, number of counted cells are indicated. CTCF indicates the corrected total cell fluorescence of images. ****$p$ value < 0.0001; $t$ test. **c** Western blot analysis using membrane protein extracts from spinal cord tissue from wild-type (WT) and MOR KO mice and anti-phospho S375/T376 MOR (AVG713-B5) antibody. **d** In vitro phosphorylation of MOR by recombinant PKCα. Purified human MOR was assembled into nanodiscs and in vitro phosphorylation was performed using recombinant PKCα. Western Blot analysis was performed using anti-phospho S363 MOR (AVG710P-B7) and anti-phospho T370 MOR (AVG711P-F2) antibodies. Blot densitometry analysis data for MOR (75 kDa) in nanodiscs was normalized to anti-Flag signal. Data represent the mean ± SD of two biological replicates. *$p$ value < 0.05; $t$ test.

been previously described as a direct substrate of PKCα[34], and our data corroborates this finding: in Neuro2A wild-type cells levels of S363 phosphorylation increased with time and remained elevated even after 60 min PMA treatment (Fig. 7a). In contrast, levels of S375/T376 phosphorylation peaked after 30 s of PMA treatment, (Fig. 7d). These increases in phosphorylation were abrogated in Neuro2A PKCα-KO cells (Fig. 7a, d), indicating that PKCα plays a role in the phosphorylation of these MOR residues.

Finally, we examined the differences in MOR phosphorylation induced by morphine or fentanyl. MOR phosphorylation at S363 did not increase after treatment with either morphine or fentanyl (Fig. 7b, c). This is consistent with previous studies that showed no effect of morphine treatment on phosphorylation at this site[33]. In contrast receptor phosphorylation at S375/T376 upon morphine treatment peaks at 1 min and returns to basal levels by 1 h (Fig. 7e, g). Interesting, the fentanyl-mediated increase in S375/T376 phosphorylation levels remained slightly elevated up to 1 h (Fig. 7f, g).

Agonist-mediated MOR phosphorylation has been reported to be modulated by distinct kinases[34]. In order to ascertain whether

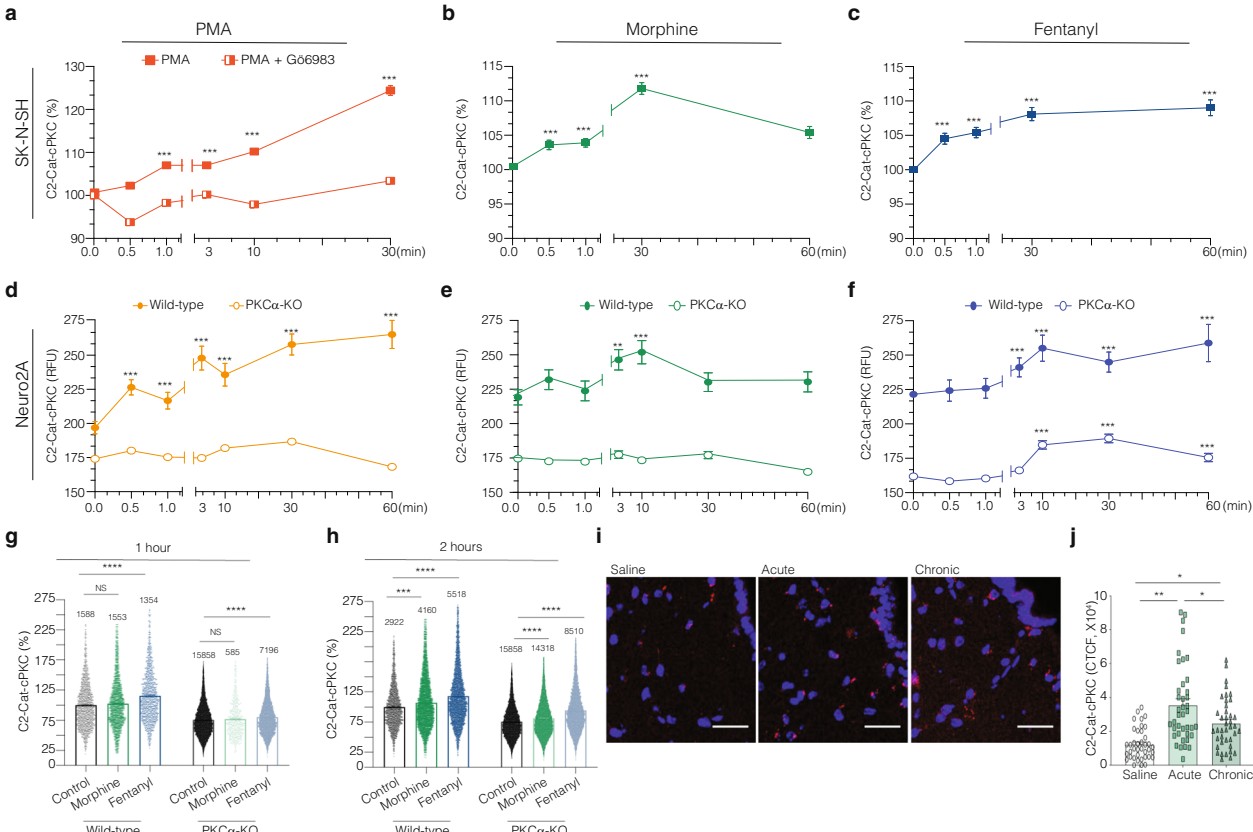

**Fig. 6 Kinetics of cPKC activation mediated by morphine and fentanyl in cell lines and mice.** Immunofluorescence analysis was carried out using the anti-C2-Cat-cPKC antibody (AVG459-D4). **a–h** Images were acquired with the InCell microscope, and the fluorescence intensity obtained using the Cell Profiler 3.1.5 software. **a–c** SK-N-SH cells were treated with PMA (500 nM) with and without Gö6983, 1 µM (**a**), morphine, 1 µM (**b**), and fentanyl, 1 µM (**c**) for different time intervals. **d–f** Neuro2A and Neuro2A PKCα-KO cells were treated with PMA, 100 nM (**d**), morphine, 1 µM (**e**), and fentanyl, 1 µM (**f**) for different time intervals. ***$p$ value < 0.001; ****$p$ value < 0.0001; One-way ANOVA. Data are mean ± SE of two experiments, $n = 400$–40,000 cells. **g**, **h** Neuro2A and Neuro2A PKCα-KO cells were treated with morphine or fentanyl for 1 h (**g**) or 2 h (**h**). Bar graph, each dot represents one cell. Lines are mean ± SE of two experiments, number of counted cells are indicated. The percentage analysis used the mean of the integrated fluorescence intensity from the non-treated Neuro2A cells as 100%. ***$p$ value < 0.001; ****$p$ value < 0.0001; One-way ANOVA. **i**, **j** cPKC activation in ventrolateral periaqueductal gray sections. Animals were administrated with saline, single injection of morphine (10 mg/kg), or chronic morphine – twice daily for 4 days. Mice were perfused with 4% paraformaldehyde for 30 min after final injection. **i** Representative images of immunofluorescence using anti-C2-Cat-cPKC antibody (red) and DAPI (blue) are shown. **j** Corrected total cell fluorescence of images in **i** were quantitated using ImageJ. Data are mean ± SE of one experiment, $n = 40$ sections/treatment, three animals each group; One-way ANOVA.

PKCα was involved in morphine- or fentanyl-mediated changes in MOR phosphorylation, we used Neuro2A PKCα-KO cells. We find that in Neuro2A PKCα-KO cells, 1 h treatment with either morphine or fentanyl causes an increase in S375/T376 phosphorylation compared to vehicle treated controls (Fig. 7g). Interestingly, in Neuro2A wild-type cells only fentanyl treatment caused S375/T376 phosphorylation at this time point (Fig. 7 g). This suggests the involvement of other kinases in Neuro2A PKCα-KO cells in MOR phosphorylation at S375/T376[44,45]. We also examined phosphorylation of T370 and T379 following 1 h treatment with morphine or fentanyl (Fig. 7h, i). In the case of T370 phosphorylation, we see that not only basal but also morphine- or fentanyl-induced levels are reduced in Neuro2A PKCα-KO compared to wild-type cells (Fig. 7h). This suggests that PKCα plays a major role in phosphorylating the T370 residue in MOR[35,36]. In the case of T379 phosphorylation, both morphine and fentanyl induce phosphorylation of this residue in Neuro2A wild-type cells. In Neuro2A PKCα-KO cells, an increase in the phosphorylation levels was only seen after morphine treatment (Fig. 7i). These data suggest that PKCα modulates T379 MOR phosphorylation only after fentanyl treatment. Differences between morphine and fentanyl-

mediated-signaling shed light over potential distinct mechanisms of receptor desensitization. Taken together, these results highlight the complexity of opioid signaling: MOR residues display distinct patterns of phosphorylation that vary both with duration of treatment and by type of agonist used.

Here we describe an integrated approach that combines a strategy of antibody development, a time- and cost-effective method of antibody validation, and a high-throughput microscopy/machine learning pipeline to identify antibodies that serve as tools to shed light on the poorly understood realm of opioid signaling. Moreover, this strategy could be broadly relevant and extended to multiple research topics besides opioid receptor signaling.

## Methods

**Antibody development.** The selection of the antigenic peptides for each selected protein was based on the following criteria: (i) sequence that shared high homology between mouse and human (Supplementary Data 3); (ii) sequence that was unique i.e., prototypic peptides not found in other proteins; (iii) sequence with more exposed hydrophobic residues; and (iv) sequence from flexible regions that do not form alpha helices or beta sheets. In order to increase the immunogenicity, additional residues were added to the antigenic sequence, if necessary. When generating phosphospecific antibodies, peptides were synthesized with non-phosphorylated

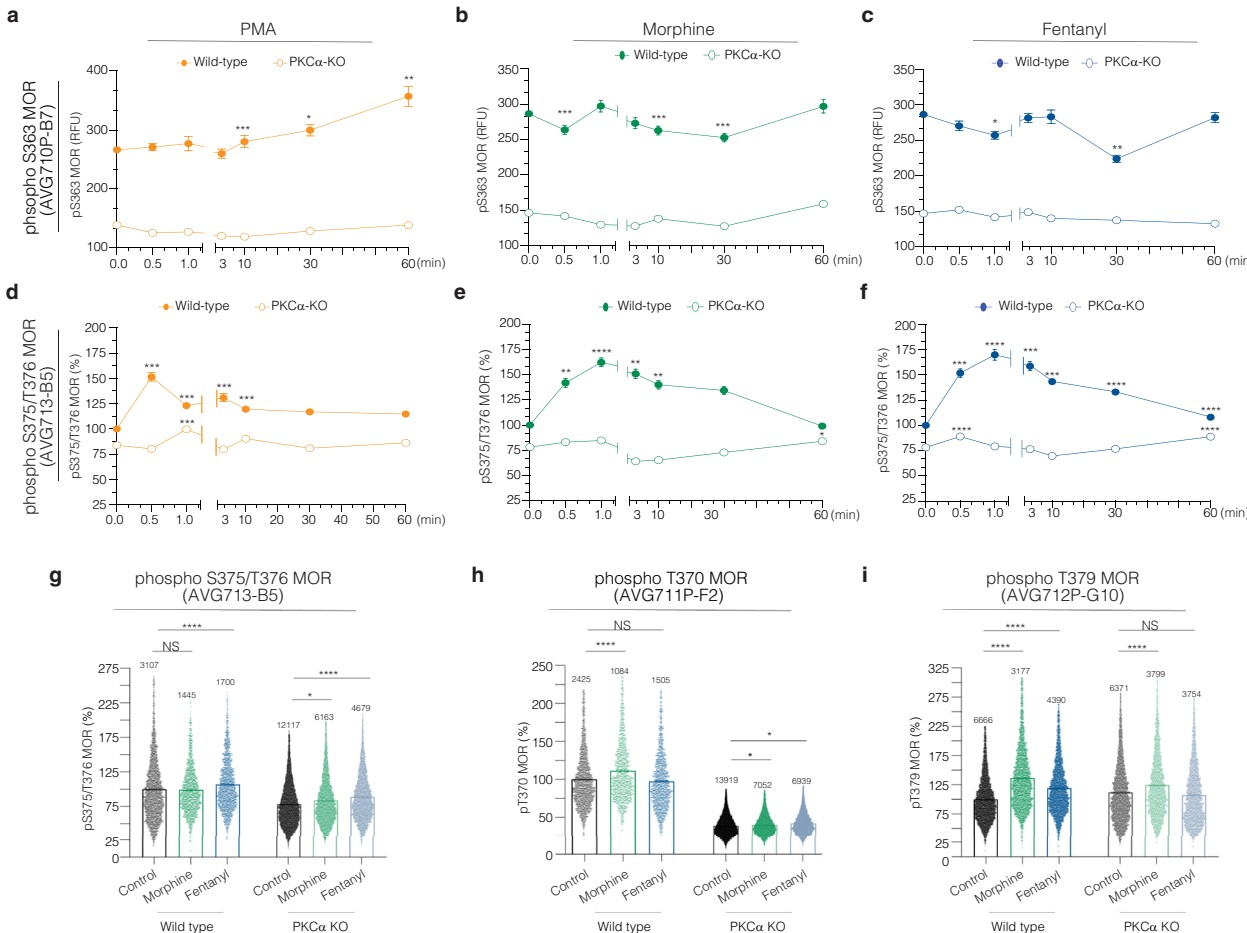

**Fig. 7 Dynamics of MOR C-terminus phosphorylation.** Neuro2A and Neuro2A PKCα-KO cells were treated with PMA (**a**, **d**), morphine (**b**, **e**), or fentanyl (**c**, **f**) for different time intervals, and probed with anti-phospho S363 MOR (AVG710P-B7) (**a**–**c**) and anti-phospho S375/T376 MOR (AVG713-B5) (**d**–**f**) antibodies. Images were acquired with the InCell microscope, and the fluorescence intensity was obtained using the CellProfiler 3.1.5 software. Data are mean ± SE of two experiments $n = 150$–3000 cells. *$p$ value < 0.01; ***$p$ value < 0.001; ****$p$ value < 0.0001; One-way ANOVA. **g**–**i** Neuro2A and Neuro2A PKCα-KO cells were treated with morphine and fentanyl for 1 h and probed using anti-phospho S375/T376 MOR (**g**), anti-phospho T370 MOR (AVG711P-B6) (**h**), and anti-phospho T379 MOR (AVG712P-G10) (**i**) antibodies. Bar graph, each dot represents one cell. Lines are mean ± SE of two experiments, number of counted cells are indicated. The percentage analysis used the mean of the integrated fluorescence intensity from the non-treated Neuro2A cells as 100%. NS non-significant; *$p$ value < 0.05; ***$p$ value < 0.001; ****$p$ value < 0.0001; One-way ANOVA.

residues, as well as their phosphorylated counterparts. If generating active state cPKC antibodies, peptides were designed from domains that become exposed following kinase activation, such as the C2-domain[22–25].

Female white New Zealand rabbits were immunized (two rabbits/ KLH conjugate) with the first immunization using 200 μg of conjugates/rabbit in Complete Freund's Adjuvant. The rabbits were further immunized with 100 μg of conjugates/rabbit in incomplete Freund's Adjuvant (booster shot) every 3 weeks for a total of 5 immunizations. After the 2nd booster shot (3rd injection), serum was harvested and tested by ELISA against BSA-conjugated peptide antigens. Wells of ELISA plates were coated with 200 ng/well of BSA conjugates or BSA only. After blocking with PBS plus 1% BSA, 1:5000 diluted serum from each rabbit was added to wells coated with BSA conjugates of the target peptide, control peptides, or no peptide. Bound antibodies were detected with 1:10,000 HRP-conjugated goat anti-rabbit antibody (Jackson Immunoresearch Inc, Cat# 111-035-006). The rabbit from each pair with the highest anti-peptide titer for a given peptide antigen was selected for a final immunization. Blood and spleen were harvested from the selected rabbit and used for library construction. All procedures were performed in accordance with Institutional Animal Care and Use Committee at R.Sargeant Animal Facility operating under NIH OLAW Assurance No. A4182-02.

For the yeast-display antibody library construction, harvested Peripheral Blood Mononuclear Cell (PBMCs) or splenocytes were used to obtain total RNA using Qiagen's RNeasy Mini Kit (Cat#74134). The total RNA was then used for cDNA synthesis. Using a specific set of primers to the heavy (HC) and light (LC) chain, antibody cDNAs were PCR amplified. The HC and LC genes were then cloned into the yeast display vector and used to transform yeast host cells. All constructed

libraries have a minimum diversity of 100 million members. Each yeast clone when induced expresses thousands of copies of a single antibody clone at its cell surface.

For the yeast display antibody library screening campaigns, one billion yeast cells from each of the constructed libraries were grown in selective medium (0.72 g amino acids without tryptophan, 7.0 g yeast nitrogen base with ammonium sulfate, 20 g glucose per liter of PBS, pH 6.5) overnight and then induced for antibody expression and display by placing the cells in induction medium (0.72 g amino acids without uracil or tryptophan, 7 g yeast nitrogen base with ammonium sulfate, 2 g glucose, 20 g raffinose, 20 g galactose per liter of PBS, pH 6.5) for 24 h. Next, 100 million of the induced cells were screened by FACS using 1 μM of biotinylated BSA-peptide conjugate and PE-labeled streptavidin. FACS was repeated for an additional 2–4 rounds with progressively lower biotinylated BSA-peptide concentrations to enrich for yeast clones expressing antibody clones that bound the target peptide.

Individual cells from the final sorting round of FACS were plated on agar plates lacking tryptophan and incubated at 30 °C for 2 days. In total, 96 colonies were picked from each yeast clone pool enriched for antigen binders and inoculated into 300 μL of selective medium in a 96 deep-well block and cultured at 30 °C overnight. On the next day 30 μL of the cell culture was transferred to a new 96 deep-well block with 300 μL of induction medium and cultured at 20 °C for 2 days.

For ELISA screening of the individual clones, 300 ng of each FITC- or BSA-peptide antigen in 50 μL of coating buffer was added to individual wells of 96-well, high-binding ELISA plates and incubated overnight at 4 °C. The next day, the wells were blocked with 200 μL of blocking buffer (PBS + 3% IgG-free BSA, pH 7.4). Next, 30 μL of binding buffer (PBS + 1% IgG-free BSA, pH 7.4) was added to each

well, followed by the addition of 20 μL of antibody-containing media supernatant from the induced yeast clone cultures. After incubation at room temperature for 1 h, the wells were washed, and bound antibodies were detected with HRP-conjugated anti-rabbit antibody. Clones that yielded strong signals in the primary ELISA were then tested in a secondary ELISA that included a panel of non-relevant peptides and protein controls.

Clones with high specificity were selected, and the DNA inserts encoding the antibody heavy and light chains were sequenced. The heavy and light chains of selected unique clones were then sub-cloned into a mammalian expression vector to be transfected in ExpiCHO cells for production as full-length rabbit IgG1 clones.

Full-length rabbit antibody expressed by the ExpiCHO cells was purified from the culture media with protein A column chromatography and buffer exchanged into PBS. The concentration of the purified antibody was determined by Lowry's assay (Pierce, Rockford IL), and the purified IgG samples used first to determine their affinity for their cognate peptide antigen compared to related or irrelevant peptide antigens, and clones with high affinity and the appropriate specificity used for further characterization.

To determine the affinity of the purified antibody, FITC- or BSA-conjugated target or control peptides were immobilized on 96-well ELISA plate to capture serial diluted purified antibody. Briefly, about 500 ng of each FITC- or BSA-conjugated peptide in 50 μL PBS was added to the wells of Immulon 2-HB ELISA plates. The plates were sealed with plastic sealing film and kept at 4 °C overnight. The next day, the wells were blocked with PBS containing 5 g/L of IgG-free BSA (ImmunoResearch, PA) for 2 h at room temperature. The plates were then washed two times with wash buffer (20 mM Hepes, 150 mM NaCl, 0.05% Triton X-100, pH 7.4). To each well, 50 μL of serial dilutions of purified antibody was added. The plates were incubated at room temperature with shaking at 145 rpm for 1 h. The wells were washed four times with wash buffer. 50 μL of HRP-conjugated goat anti-rabbit IgG (1:5000) was then added to each well and incubated for 1 h at RT, followed by four washes with wash buffer. In total, 50 μL of HRP substrate, TMB (KPL, MD), was then added to each well and incubated for 10 min. The reaction was stopped by adding 50 μL 100 mM HCl and the absorbance values at OD 450 nm were determined with a plate reader (Molecular Devices, CA). Prism software (GraphPad, San Diego, CA) was used to plot the data and perform curve fitting for a one site binding curve to determine apparent $K_D$ values.

**Cell culture and genome editing**. Neuro2A, SK-N-SH and HEK293 cell lines were obtained from ATCC and cultured with DMEM (Gibco) supplemented with 10% fetal bovine serum (FBS, Gibco) and 1% penicillin-streptomycin (Sigma) in a humidified 5.0% $CO_2$ atmosphere at 37 °C. Knockout cell lines to PRKCA, PRKCB and OPRM genes were generated using CRISPR/Cas 9 system. The single guide RNAs (gRNA) were designed using the online predictor CCTop-CRISPR/Cas9 target (http://crispr.cos.uni-heidelberg.de)[46,47]. To identify the optimal gRNA for each gene, we used the following criteria: (i) the exonic region of interest should target all the potential gene isoforms; (ii) the target region was near the 5′ UTR start of the coding sequence; (iii) the selected gRNA presented the minimum number of off-targets; (iv) all the off-targets were located on an intergenic, intronic or non-coding region; (v) all the off-targets contained at least two nucleotide differences within the seeding region of the gRNA sequence; (vi) all the off-targets presented a distance of at least two nucleotides from the PAM sequence. The gRNAs (Table 1) were cloned in the PX459 V.2 plasmid (#48139 Addgene) as described by Ran et al.[48].

Neuro2A cells were transiently transfected with PX459 V.2 plasmid containing the gRNA and Cas9 using Turbofectamine® (Invitrogen), according to the manufacturer's protocol. After 48 h, the cells were expanded into 10 cm² dishes and puromycin (2 μg/mL) was added for additional 48–72 h, in order to obtain complete selection. The antibiotic was washed out and the cells were cultivated for further experiments (not exceeding two or three passages). The gRNA efficiency was tested by genomic PCR, the specific set of primers for each genomic region is shown on the Table 1.

For the Neuro2A expressing MOR, the transfection protocol was performed as previous described. MOR was N-terminally tagged with a pH-sensitive GFP (SpH) and the plasmids SpH-MOR, SpH-MOR S363A, and SpH-MOR T370A were a kind gift from M. Puthenveedu[28]. After 48 h of transfection, the cells were fixed and the immunofluorescence staining was performed as described below.

**Cell treatment**. Cells were seeded in 96-well plates (High-content; cat#4517; Corning) at 25,000 cells per well. After 24 h the cells were treated with PMA (100 nM), with and without Gö6983 (1 μM), morphine (1 μM), or fentanyl (1 μM).

To explore the temporal dynamics, we treated the cells for 30 s, 1, 3, 10, 30, or 60 min starting with the latest time point first. All the cells in the plate were fixed at the same time using 4% PFA as described below.

**Immunofluorescence and high-content microscopy analysis**. For the validation screening of the rAbs, Neuro2A cells were fixed with 4% PFA at room temparture for 10 min, washed with 1x PBS, followed by treatment with 100% methanol at −20°C for 5 min. The cells were permeabilized with PBS, 0.05% saponin, and 3% BSA, followed by incubation with the antibodies of interest, for 16 h at 4 °C The unbound antibody was washed off using PBS and 0.05% saponin buffer, followed by incubation with secondary antibodies at room temperature, for 1 h (protected from light) and removal of unbound antibody. The images were acquired using the high-throughput microscope (InCell, GE Healthcare) with a ×40 objective. For each replicate we acquired 5–10 fields per well.

For the kinetic experiments, the cells were fixed with 4% PFA at room temperature for 15 min. Fixed cells were blocked with PBS containing 0.1% Triton-X100, and 3% BSA; the subsequent steps for antibody staining were similar as described above.

The fluorescence intensity measurement was performed using the Cell Profiler 3.1.5[11]. Since the Cell Profiler recognizes single cells, the immunofluorescence images cannot contain high background and/or unspecific staining. In addition, the cells need to be in a monolayer, avoiding cell clusters. Our pipeline was designed to recognize first the nucleus (stained by DAPI), followed by the immunofluorescent signal surrounding the entire cell. This strategy allows us to properly identifiy the cells and measure changes in protein expression in the KO cells. Finally, we ensured that the automated Cell Profiler recognition is capturing the correct cells by performing a visual examination of the output data and ensuring that only cells are highlighted.

The Cell Profiler parameters were: (i) Identify the Primary Object, selecting DAPI as input image. The typical diameter of objects, in pixel units, was Min 50–Max 500. We used adaptative as threshold strategy, Otsu as thresholding method, and two-class thresholding. The threshold smoothing scale was 1.2. The threshold correction factor was 1.0, and the size of adaptive window was 50. We used shape as a method to distinguish clumped objects, and intensity as the method to draw dividing lines between clumped objects. (ii) Identify the Secondary Object, by selecting rAbs as input images, and DAPI as input objects. We used propagation as selected method to identify the secondary objects, adaptative as threshold strategy, Otsu as threshold method, and three-class thresholding. Assigned pixels in the middle intensity class to the foreground. The threshold smoothing scale was 0.005. The threshold correction factor was 1.08. The size of adaptive window was 250, with regularization factor of 0.5. We selected to fill hole in identified object and to discard secondary objects touching the border of the image. (iii) Measure object intensity, selecting the rAbs as input image and the secondary objects as objects to measure.

**Animal subjects and experimental treatment**. Male C57BL/6NCrl mice (Charles River, CA) weighing 25–30 g at 6–8 weeks old were used in this study ($n = 3$ per group). Mice were group-housed (5 per cage) in a humidity and temperature-controlled room with a 12 h light/dark cycle (07:00–19:00 h). Morphine or saline administration took place between 08:00 and 16:00 h. All procedures were performed in accordance with Institutional Animal Care and Use Committee at Utah State University, Protocol 2775. Animals in the chronic morphine group were administered 10 mg/kg of morphine sulfate in saline (West-Ward Pharmaceuticals) twice daily for 4 days. On Day 5, a final injection was given followed by transcardial perfusion 30 min later. Animals in the acute morphine group received only one administration of 10 mg/kg morphine 30 min prior to perfusion. Animals in the saline group received one dose of 10 ml/kg of 0.9% saline 30 min prior to perfusion.

**Immunohistochemistry**. Mice were deeply anesthetized with isofluorane (99.9% inhaled) and transcardially perfused through the ascending aorta with 4% paraformaldehyde (100 mL). Brains were postfixed for 1 h and stored in 1× PBS. Immunohistochemistry was performed on free-floating coronal brain slices (50 μm) containing the periaqueductal gray. Sections were incubated in 1% sodium borohydride in PBS for 30 min followed by blocking buffer (5% normal goat serum and 0.3% Triton X-100 in PBS) for 1 h. Tissue was incubated overnight at 4 °C in primary antibody C2-Cat-cPKC (rabbit, 1:500) in 1% BSA and 0.1% Triton X-100. The primary antibody was visualized with goat anti-rabbit A594

| Gene | gRNA sequence | Primer validation (Fw) | Primer validation (RV) |
|---|---|---|---|
| PRKC alpha | GCTTGTCCGGGTGCGGATAA | CTCGCCCATTGTGAACTGTA | GATGGAACCTGGGTCTTCTAAC |
| PRKC beta | CGGTGCGGACAAGGGCCCG | GTGTGTATGTGTGTGTGTATG | GACTTCCCAGCAAAGGACTAA |
| OPRM | GTCGGGGGGTACGGAAATCC | GAAGACTGCCACCAACATCTA | AGGCAACTGCTTGGAAATTATG |

**Table 1 gRNA sequence and genomic PCR primers.**

(Life Technologies, A11037). After 5 min of incubation with DAPI (100 ng/ml), sections were mounted with ProLong Gold Antifade (Molecular Probes). Images were taken with a Zeiss LSM 710 confocal microscope at the Microscopy Core at Utah State University. Quantification of PKC-fluorescence was performed using the ImageJ integrated density measurement on thresholded images. Cells in the field of view were randomly selected along with areas representing background fluorescence. The fluorescence from anti-C2-Cat-cPKC signal was calculated using the formula for Corrected Total Cell Fluorescence (Integrated Density–(area of selection × mean background fluorescence)) (McCloy, 2014). CTCF values of the three groups were compared using unpaired t-test on Graphpad Prism (version 7). Data are represented as mean ± SEM. The experimenter was not blinded to treatment conditions.

**Western blot**. The total protein extraction was performed using a non-denaturing lysis buffer (50 mM Tris, 120 mM NaCl, 5 mM EDTA, 50 mM NaF, 0.5% NP-40, phosphatase and protease inhibitors), or a denaturing lysis buffer (RIPA buffer, 150 mM NaCl, 5 mM EDTA, 50 mM Tris, 1% NP-40, 0.5% sodium deoxycholate, 0.1% SDS, phosphatase and protease inhibitors) and the synaptosome fractionation was perfomed as described by Morón et al.[49]. The protein samples were solubilized with 4x loading buffer at 60°C for 20 min. 12.5% separating gels were prepared for proteins with MW < 40 kDa and 10% separating gels were prepared for proteins with MW > 40 kDa. The gels were run at 130 V for 2 h using electrophoresis buffer (Tris 8 g, Glycine 14.4 g, 10% SDS 10 ml and $H_2O$ up to 1000 mL). The gels were further placed in a transfer sandwich holder with nitrocellulose membrane, filter paper and sponges pre-soaked with 1x Transfer buffer (Tris 4.04 g, Glycine 14.42 g, Methanol 200 ml and $H_2O$ up to 800 mL). The proteins were transferred to nitrocellulose membrane at 110 V for 1.5 h for high molecular weight proteins (40 kDa or more) and at 0.3 A for 30 mins for low molecular weight proteins (>40 kDa). The dye was washed off completely with 1x Tris Buffered Saline Tween-20 (TBSt; 100 mL of 10x TBS (Tris 6.06 g, NaCl 21.41 g), 0.5 ml of Tween 20 and $H_2O$ up to 1000 mL) solution. Further, the membranes were blocked for 30 mins at 37 °C with Odyssey Blocking Buffer (LI-COR biosystems) and incubated with primary antibody (1:500 in Odyssey Blocking Buffer) for 24 h on a platform shaker at 0–4 °C. Unbound primary Ab was removed by washing the membranes 3x times with 1xTBST buffer. Each wash was performed for 15 mins on a platform shaker at 0–4 °C. The membranes were incubated with secondary Ab (1:1000 in Odyssey Blocking Buffer; Odyssey IR Imaging system Goat anti-rabbit IR Dye 680 (red channel) or 860 (green channel), LI-COR biosystems) for 30 mins on a platform shaker at 4 °C (protected from light). Unbound secondary Ab was removed by washing the membrane 3x times with 1xTBST buffer. Each wash was performed for 15 mins on a platform shaker at 0–4 °C. The membranes were analyzed using Odyssey classic ODY-1076, LI-COR biosystems.

**ELISA**. Protein was isolated by homogenization of HEK293 and Neuro2A cells using lysis buffer (50 mM Tris, 120 mM NaCl, 5 mM EDTA, 50 mM NaF, 0.5% NP-40, phosphatase, and protease inhibitors). The technical triplicates were plated containing 20 µg protein from respective cell lines in a 96-well plate (cat#3690, Costar). After 24 h of incubation at 37 °C, wells were washed three times with Tris-buffered saline with Tween20 (TBSt, 20 mM Tris, 150 mM NaCl, 0.1% (w/v) Tween20). Wells were blocked with 1% BSA in TBSt for 1 h at room temperature (RT) and washed again with 1xTBST buffer. Wells were then incubated with primary Ab (1:200 in 3% BSA) for 24 h at 0–4 °C. Unbound primary Ab was washed with TBSt buffer. Peroxidase labeled anti-rabbit IgG secondary Ab (1:1000 in 3% BSA; Vector Labs, Lot#ZC0304) was added to the wells and the plates were incubated for 1 h at RT. Unbound secondary Ab was washed with TBSt buffer. In all, 100 µL of freshly made peroxidase substrate (5 mg OPD [o-Phenylenediamine] in 0.1 M citrate phosphate buffer (pH 5.0) [0.1 M citric acid (A) and 0.2 M $Na_2HPO_4$ (B). Take 12.2 mL (A) + 12.8 mL (B). Adjust volume with $H_2O$ up to 50 mL]) and 10 µL $H_2O_2$ were added per well (protected from light). The reaction was stopped after 5 mins with 50 µL of 5 N $H_2SO_4$ as the quencher. The colorimetric reaction was read using SpectroMax M3 at 490 nm optical density and analyzed with SoftMax Pro 7.0.3. The ELISA was carried out using anti-Flag (Sigma, #F1804), anti-Myc (Santa Cruz, #SC-40), anti-HA (Invitrogen, #26183) and the antibodies listed in the Supplementary Data 1 and Table 1.

**MOR purification and in vitro phosphorylation by PKCα**. MOR purification was performed as described by Kuszak et al.[30] and Che et al.[50]. High-titer of MOR recombinant baculovirus was generated using Bac-to-Bac Baculovirus expression system (Invitrogen). Expression of MOR was carried out by infection of SF9 cells ($3.5 \times 10^6$ cells/mL) in ESF921 media (Expression Systems) at a MOI (multiplicity of infection) of 2. Naltrexone at 1 µM final concentration was added and after 48 h the cells were counted and harvested by centrifugation. The cell viability should be in the range of 90%.

Membrane protein purification was performed by washing the cell pellets in a low-salt buffer [10 mM HEPES, pH 7.5, 10 mM $MgCl_2$, 20 mM KCl, and protease inhibitors (Roche)]. To remove soluble and membrane-associated proteins, the protein pellet was washed four times using a high osmolarity buffer [1.0 M NaCl,

10 mM HEPES, pH 7.5, 10 mM $MgCl_2$, 20 mM KCl, and protease inhibitors (Roche)]. The centrifugation was performed at $40,000 \times g$ for 30 min at 4 °C.

MOR purification was performed by resuspending the membrane pellets in resuspention buffer [25 mM HEPES, 150 mM NaCl, 10 mM $MgCl_2$, 20 mM KCl, 50 µM naltrexone, and protease inhibitors (Roche)]. After 1 h incubation at room temperature, followed by 2 h incubation at 4 °C the supernatant was collected following centrifugation at $150,000 \times g$ for 30 min at 4 °C The next steps were performed at 4 °C. A solution containing 20 mM imidazole, 800 mM NaCl, and TALON IMAC resin (Clontech) was added to the supernatant, followed by agitation for 90 min. After spinning at 700 g for 5 min, the resin was washed with 10 colum volumes (cv) of wash buffer I [25 mM HEPES, pH 7.5 containing 500 mM NaCl, 0.01% (w/v) LMNG, 50 µM naltrexone, 20 mM Iimidazole, 10% (v/v) glycerol], followed by 10 cv of wash buffer II [25 mM HEPES, pH 7.5 containing 500 mM NaCl, 0.05% (w/v) LMNG, 50 µM naltrexone, 10% (v/v) glycerol]. Proteins were eluted in 2.5 cv of Wash Buffer II + 250 mM imidazole, concentrated in a 100-kDa molecular weight cut-off Vivaspin 20 concentrator (Sartorius Stedim) and imidazole was removed by desalting the protein over PD MiniTrap G-25 columns (GE Healthcare).

The purified MOR was assembled into nanodiscs using lipids and MSP1D1, as the scaffold protein[31]. The lipid components were porcine polar brain lipid extract (Avanti Polar Lipids), 1-palmitoyl-2-oleoyl-sn-glycero-3-phosphocholine (POPC), and 1-palmitoyl-2-oleoyl-sn-glycero-3-[phospho-rac-(1-glycerol)] (POPG) at a final concentration of 7 mM lipids in a molar ratio of 1.07:1.5:1 brain lipid:POPC: POPG. Lipids, purified MOR, and MS1P1D1 were solubilized in homogenization buffer [20 mM Hepes, pH 8.0 containing 100 mM NaCl, 1 mM EDTA, and 40 mM sodium cholate] using a molar ratio of 1:4:320, MOR:MSP1D1:lipid. After 2 h of incubation at 4 °C, the detergent was removed using Bio-BeadsTM (Bio-Rad, 0.05 mg/ml of reconstitution volume).

The protocol for in vitro phosphorylation of MOR by PKCα was adapted from Cenni et al.[51]. Classical PKCs, such as PKCα, are activated by $Ca^{2+}$, lipids, phosphatidylserine (PS), and diacylglycerol (DAG). PS (2800 nmol) and DAG (76 nmol) were solubilized in $CHCl_3$, and mixed in a glass tube. The solvent was evaporated under a stream of nitrogen. Next, the lipids were resuspended in 2 mL 20 mM HEPES pH 7.4 by vigorous vortexing, and sonication. The kinase assay was performed using MOR (100 pmol) and recombinant PKCα (10 pmol) in Go buffer [20 mM HEPES, pH 7.4 containing 1 mM DTT, 5 mM $MgCl_2$, 100 µM ATP, 140 µM/3.8 µM PS/DAG, and 100 µM $Ca^{2+}$]. After 10 min at room temperature, the reaction was stopped using loading buffer and the Western blot analysis was performed.

**Statistics and reproducibility**. Graphical illustrations are plotted using Prism software (GraphPad, San Diego, CA). The *P*-value cut-off used in this study is 0.05. *, **, ***, and **** in the figures refer to *P*-values ≤0.05, ≤0.01, ≤0.001, and ≤0.0001, respectively. The statistical analysis and the sample size are described in the figure legends. We considered biological replicates the measurements of biologically distinct samples.

**Reporting summary**. Further information on research design is available in the Nature Research Reporting Summary linked to this article.

## Data availability
The data that support the findings of this study are available from the corresponding author upon reasonable request.

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

## Acknowledgements

We acknowledge Emily Chao, Kenix Vo, Jason Higa M.S., and Donghui Li, M.S. of AvantGen for their technical contributions to the screening, sequencing and antibody clone purification of all the antibody clones. We would like to thank Dr. Daniel Wacker, Dr. Michael J. Capper, Jeshua de Tenorio, Gregory Zilberg, and Audrey L. Warren for the technical support in the opioid receptor purification and its assembly into nanodiscs. We would like to thank Chenge Liu and Andrei Jeltyi for their help with the western blots. We would like to thank Alan Stern for the technical assistance using InCell microscope and Seshat Mack for critical reading of the manuscript. We would like to thank Joel Glovier for authorizing the use of the Erlenmeyer drawing in Fig. 2. This work was supported by NIH grants, 1R44DA35531 (to X.F.) and DA008863 and NS026880 (to L.A.D.), and Yale/NIDA Neuroproteomics Center Grant 5P30DA018343 (PI, Marina R. Picciotto).

## Author contributions

Mariana Lemos Duarte planed the experiments, developed the CRISPR/Cas9 knock-out cells, validated the antibodies, developed the HTM/machine learning protocol, performed the kinetic assay and wrote the manuscript. Nikita Trimbake and Achla Gupta helped to validate the antibodies. Xiaomin Fan is the inventor of the yeast antibody display technology used for antibody discovery and isolation. He designed the peptide sequences for each of the protein antigens selected for the project and had overall oversight of the project. Catherine Woods served as project manager and oversaw the project from rabbit library screening to antibody production, generated and oversaw the rAb database for the final set of selected clones, proofed the sequences, was involved in clone production characterization from start to finish, and served as the liaison between AvantGen and Mount Sinai. Christine Tumanut produced the rabbit antibody libraries from blood of sero-positive rabbits, screened the libraries and coordinated ELISA analysis of individual clones as well as the internal rabbit rAb database for tracking the panels of rAb clones evaluated during the project. Akila Ram and Erin Bobeck performed the animal experiments and immunofluoresce analysis using brain slices. Ivone Gomes helped with the writing of the Results and Discussion sections. Deborah Schechtman advised on the design of the PKC antibody epitopes, helped with the results, discussion, and editing of the manuscript. Lakshmi Devi served as the PI for the project and was involved in the design of the studies, interpretation, and presentation of the data and writing of the manuscript.

## Competing interests

The authors declare no competing interests.
