## [Peer Review File · Communications Biology]

Reviewers' comments:

Reviewer #1 (Remarks to the Author):

This manuscript describes the development of a panel of antibodies relevant to opioid signalling and their application in immunofluorescence studies of such signalling in Neuro2A cells. The antibody development seems well done, as far as antibodies go, but I am constantly amazed at how antibodies' specificity is over-sold. Some concerns:

1. "Grammable"? Really? Please change the title.
2. Abstract: the abstract over-sells what this manuscript does towards being able to detect changes in protein conformation or PTMs. As far as I could tell, they only developed one Ab for detecting conformation (the activated PKC one) and a few phospho-specific ones. Conformational reagents are pretty rare so this is a good achievement but the abstract suggests that we're going to be reading about a lot of these. This should be reworded
3. Specificity. Antibodies just aren't as specific as is often sold. As an example, there are multiple bands on most of the Western blots shown. Some are hilariously bad (e.g., Ap2b1). These could be isoforms but they're more likely other proteins with some affinity for the Ab. I would like to see more discussion of the caveats around following the gold-standard approaches for developing and validating antibodies yet there still being the potential for non-specific binding
4. HTM artifacts? I am intrigued by the violin plots in figure 2. Why is there apparent periodicity in them? Is there some kind of systematic effect in the automated quantitation?
5. Antibody availability. How are these reagents going to be made available to others? At the very least it seems that the hybridomas should be deposited into one of the public banks. The authors simply offering to make them available upon request is not acceptable.

Reviewer #2 (Remarks to the Author):

In this manuscript, the authors generate and initially characterise a large number of rabbit monoclonal antibodies. It remains unclear to this reviewer how this work directly relates to the opioid crises. In this regard, the characterisation of phosphorylation-specific opioid receptor antibodies is preliminary at most. Shown are higher levels of cytoplasmic fluorescence in μ -receptor expressing cells compared to WT cells. One would expect that the receptor is labeled at the plasma membrane at least shortly after activation. In addition, the authors fail to provide any Western blot data that would lend more confidence to the specificity of the phospho- μ antibodies.

Reviewer #3 (Remarks to the Author):

An impressive effort for generating high-quality antibodies to synaptic and cell signalling components and post-translational modifications.

Comments and Suggestions

1. The title does not fully describe the manuscript, which is the generation and validation of rabbit monoclonal antibodies (mAbs) to synaptic proteins that likely play a role in opioid cell signaling. Nor is the term "grammable" clear or explained in the Abstract.
2. The Abstract may wish to provide more experimental details: how were the mAbs generated and validated, what is their specificity and affinity, and if any new biology was learned. Presumably peptides corresponding to human proteins served as targets. Was cross-reactivity between mouse and human proteins part of the grand plan?

3. A few questions come to mind reading the manuscript. How many of the mAbs still reacted with an antigen in an engineered KO cell? What is the quantitative improvement in time and cost efficiency of the high-throughput microscopy pipeline? Any negatives or limitations to the pipeline? What residues were phosphorylated in the Mu-opioid receptor? Do such antibodies cross-react with the non-phosphorylated peptides or receptor? Do the mAb only recognize linear epitopes? Do the hybridomas secrete more than one mAb?
4. How did you select the antigenic peptide of the C2 domain of PKC, which would only be accessible in active, open PKC?
5. What percentage of mAbs failed to work in immunohistochemistry or immunofluorescence staining? Presumably not all epitopes are accessible in paraformaldehyde-fixed samples for a variety of reasons.
6. Any guess of what protein kinase is responsible for phosphorylating T379 of MOPr?
7. It is unclear how many proteins served as antigens for this study, as much of the description focuses on just a few. Highlighting what proteins in a pathway (cartoon) had mAb generated and validated might be helpful to the reader.
8. Has the coding sequences of the best mAbs been sequenced? This would ensure renewability.

We thank the reviewers for their feedback and for their very helpful suggestions. We have carried out additional studies and included the new data in the revised manuscript. This resulted in the strengthening of the findings in the manuscript. A point-by-point response to the reviewers' concerns are detailed below.

Reviewers' comments:

Reviewer #1:

This manuscript describes the development of a panel of antibodies relevant to opioid signalling and their application in immunofluorescence studies of such signalling in Neuro2A cells. The antibody development seems well done, as far as antibodies go, but I am constantly amazed at how antibodies' specificity is over-sold.

We agree that an increased focus on the validation of the antibodies would provide more value to the manuscript. In order to provide more information about the antibodies' specificity, we carried out the following additional experiments:

1. To improve the quality of the data, we optimized our Western Blotting (WB) and our immunocytochemistry (ICC) protocols. We re-analyzed 114 antibodies by WB and repeated the high-throughput microscopy (HTM) to screen 133 antibodies. The new data is now included as revised Figures 2 - 5, as well as Supplementary Figures 1, 2, 4, 5, 6, 8 and 9. All the data are summarized in Supplementary Tables 1 and 3.
2. Due to our interest in opioid receptor signaling, we evaluated the specificity of some of these antibodies using four complementary approaches: (i) CRISPR/Cas9 to knock out (KO) the genes for MOR, PKC α or PKC β in cells; (ii) using membrane protein extracts from spinal cord tissue of MOR KO mice, (iii) using phospho-site specific MOR mutant cells; and (iv) by carrying out *in vitro* phosphorylation of purified MOR. The new data supporting the specificity of these antibodies are included in revised Figures 3 - 5 and Supplementary Figures 8 - 10.

1. "Grammable"? Really? Please change the title

We have modified the title to *High throughput functional screening for the development and validation of antibodies against synaptic proteins: A focus on opioid receptor signaling.*

2. Abstract: the abstract over-sells what this manuscript does towards being able to detect changes in protein conformation or PTMs. As far as I could tell, they only developed one Ab for detecting conformation (the activated PKC one) and a few phospho-specific ones. Conformational reagents are pretty rare, so this is a good achievement, but the abstract suggests that we're going to be reading about a lot of these. This should be reworded.

We have reworded the abstract.

3. Specificity. Antibodies just aren't as specific as is often sold. As an example, there are multiple bands on most of the Western blots shown. Some are hilariously bad (e.g., Ap2b1). These could be isoforms but they're more likely other proteins with some affinity for the Ab. I would like to see more discussion of the caveats around following the gold-standard approaches for developing and validating antibodies yet there still being the potential for non-specific binding.

The reviewer makes an excellent point. We have added a detailed discussion about the affinity of the antibodies in the Results section (Pages 3 and 4). We had previously tested a subset of the antibodies under non-denaturing conditions. To address the reviewer's concerns, we optimized the protein extraction protocol to include a denaturing lysis buffer and analyzed 114 antibodies by WB (Figure 2, Supplementary Figure 1 and Supplementary Table 1). In addition, we tested 133 antibodies by HTM using buffer containing saponin that reveals better localization for the synaptic proteins (Supplementary Figures 5 and 6, and Supplementary Tables 1 and 3). The importance of exploring different strategies to perform the ICCs is described in Supplementary Figure 4. The criteria to test the efficacy of the antibodies in HTM analysis is

illustrated in Figure 3. We considered the antibody to be selective if it revealed a difference in fluorescence intensity that changed with the antibody concentrations (1:500; 1:100) and exhibited subcellular localization that matched the localization reported in the literature. Finally, we tested the specificity of a subset of antibodies using knock-in and knock-out strategies or by using antigenic site mutant proteins (Figures 3 - 5, Supplementary Figures 3, 8, 9 and 10).

4. HTM artifacts? I am intrigued by the violin plots in figure 2. Why is there apparent periodicity in them? Is there some kind of systematic effect in the automated quantitation?

To address this concern, we revised the analysis; this analysis was performed on Cell Profiler (pipeline provided in the Supplementary File 1) and the data was plotted using GraphPad Prism. In addition, correlation analysis was performed comparing the Cell Profiler results with ImageJ (Supplementary Figure 7). The periodicity of the shape in violin plots remains in this revised analysis.

5. Antibody availability. How are these reagents going to be made available to others? At the very least it seems that the hybridomas should be deposited into one of the public banks. The authors simply offering to make them available upon request is not acceptable.

The antibody clones were isolated by contract # R44DA035531 from NIDA. As part of the SBIR program, our small business collaborator is expected to commercialize the antibody clones so that they are available to the neuroscience research community. They will provide access to the clones through a portal on their company website following final acceptance of the manuscript. Due to the blinded nature of this review, we have withheld the company's name (and are happy to provide it if the editor requests it).

Reviewer #2:

In this manuscript, the authors generate and initially characterise a large number of rabbit monoclonal antibodies. It remains unclear to this reviewer how this work directly relates to the opioid crises. In this regard, the characterisation of phosphorylation-specific opioid receptor antibodies is preliminary at most. Shown are higher levels of cytoplasmic fluorescence in m-receptor expressing cells compared to WT cells. One would expect that the receptor is labeled at the plasma membrane at least shortly after activation. In addition, the authors fail to provide any Western blot data that would lend more confidence to the specificity of the phospho-?? antibodies.

We would like to thank the reviewers for the comments. With regards to the characterization of phosphospecific antibodies, our previous ICC analysis was performed using 0.05% Triton X (Methods), a detergent which disrupts the structure of the plasma membrane and, consequently, the location of the membrane proteins. In order to reveal better subcellular localization, we used 0.1% saponin in the blocking buffer (Figure 5b and Supplementary Figures 5 and 6). The justification for using different strategies to perform the different immunocytochemistry protocols is described on Supplementary Figure 4.

To test the selectivity of the phospho-MOR antibodies, we used three different approaches: (i) *in vitro* phosphorylation of purified MOR by PKC α ; (ii) overexpression of MOR C-terminal mutants in Neuro2A cells; (iii) membrane protein from spinal cord tissue of MOR KO mice. We find a lack of signal in the phosphorylation site mutants (by HTM) and MOR KO mouse brain tissues (by WB), as well as an increase of signal on the MOR phosphorylated residues after *in vitro* phosphorylation by PKC α (by WB). Substantial amounts of new data showing the specificity of the phospho-specific antibody are included as new figure 5.

Reviewer #3:

An impressive effort for generating high-quality antibodies to synaptic and cell signalling components and post-translational modifications.

We would like to thank the reviewer for acknowledging our efforts.

Comments and Suggestions

1. The title does not fully describe the manuscript, which is the generation and validation of rabbit monoclonal antibodies (mAbs) to synaptic proteins that likely play a role in opioid cell signaling. Nor is the term "grammable" clear or explained in the Abstract.

We have changed the title.

2. The Abstract may wish to provide more experimental details: how were the mAbs generated and validated, what is their specificity and affinity, and if any new biology was learned. Presumably peptides corresponding to human proteins served as targets. Was cross-reactivity between mouse and human proteins part of the grand plan?

The abstract has been reworded to include the suggestion by the reviewer.

3. A few questions come to mind reading the manuscript.

How many of the mAbs still reacted with an antigen in an engineered KO cell?

We tested seventeen antibodies using HTM and 14 did not give a signal (only three gave signal) in KO cells (Figures 3 - 5, Supplementary Figures 9 and 10). In addition, we used PKC α KO cells to investigate changes in the phosphorylation states of three of its substrates, PP2A, MARCKS and MOR (Figure 3); as expected in PKC α KO cells a substantial loss of phosphorylation is seen only at the site known to be phosphorylated by this enzyme (Figure 3 and Supplementary Fig 10).

What is the quantitative improvement in time and cost efficiency of the high-throughput microscopy pipeline?

HTM analysis was performed using 96-Well plate optimized for fluorescence and luminescence in cell culture and microscopic applications (Thermo Cat #152037, case with 20 plates the price is \$385.00). The image acquisition was performed using the IN-Cell Analyzer Microscope (Ge Healthcare). The incubation was performed using 50 μ L of antibody solution per well. Since the antibody dilution used in the analysis was 1:500, to probe all the 96-wells less than 10 μ L of mAbs was necessary. Finally, the analysis is performed using Cell Profiler, an open-source software provided by the Broad Institute.

To carry our immunocytochemistry, we need 48 hours. The image acquisition is performed in 2 – 4 hours (depending on the number fields acquired per well). The analysis of one 96-well plate using Cell Profiler software takes 4 – 6 hours to be completed.

Any negatives or limitations to the pipeline?

The reviewer raises an excellent point. Since the Cell Profiler software recognizes single cells, the immunofluorescence images should not contain high background and/or unspecific staining. In addition, the cells need to be in a monolayer, avoiding cell clusters. Our pipeline was designed to recognize first the nucleus (stained by DAPI), followed by the immunofluorescent signal of the entire surrounding cell. This strategy allows us to properly identify the cells and measure changes in protein expression in the KO cells.

Finally, we ensure that the automated Cell Profiler recognition is capturing the correct cells by performing a visual examination of the output data and ensuring that only cells are highlighted. We have included these further details in the Methods section.

What residues were phosphorylated in the Mu-opioid receptor? Do such antibodies cross-react with the non-phosphorylated peptides or receptor?

Phosphorylation of the C-terminus of MOR has been described to play key roles in MOR desensitization and internalization (reviewed in Lemos Duarte and Devi, 2020). The phosphorylated residues in MOR are: Y336; S355; T357; S363; T370; S375; T376 and T379. The affinity of the phospho-specific antibodies to the non-phosphorylated peptides is described in Supplementary Table 1.

Do the mAb only recognize linear epitopes?

We know that during the preliminary testing at our small business collaborator, the antibody clones recognized their target linear peptide. The studies conducted in my laboratory at Icahn School of Medicine at Mount Sinai evaluated the clones with lysates or membrane preparations containing native protein. So, while we can say the clones recognize their cognate peptide antigen within the context of the intact target protein conformation, we cannot confirm that they recognize/bind to a non-linear conformation of the epitope.

Do the hybridomas secrete more than one mAb?

The antibody clones are isolated from yeast display antibody libraries constructed using antibody genes isolated from B cells obtained from immunized rabbits. Clones that are found to be specific in primary and secondary ELISAs using the target antigen alongside non-relevant proteins are sequenced and the heavy and light chains of the Fab subcloned into a full-length IgG1 vector for production as purified IgG1s from ExpiCHO cells cultures. Therefore, each IgG arises from a well-defined pair of heavy and light chain sequences so unlike hybridoma technology there is no possibility of more than one clone being secreted by the transfected ExpiCHO cells.

4. How did you select the antigenic peptide of the C2 domain of PKC, which would only be accessible in active, open PKC?

The rationale is that during the initial step in the activation of PKC, a conformational change in the enzyme leads to anchoring of the kinase to the membrane fraction; this, in turn, leads to the exposure of the C2 domain making it accessible for interaction with substrate (Antal et al. 2015). This information was previously used in the development of conformational specific antibodies against the C2 domain of the PKC (Pena et al. 2016). In the present study, we used the same antigenic sequence to develop rabbit monoclonal antibodies that will be available to the scientific community.

5. What percentage of mAbs failed to work in immunohistochemistry or immunofluorescence staining? Presumably not all epitopes are accessible in paraformaldehyde fixed samples for a variety of reasons.

The reviewer makes an excellent point. We recently rescreened all of the antibodies by WB and ICC (Figures 2, 3 and Supplementary Figures 1, 2, 5, 6 and 9). To measure the efficiency of the antibodies in HTM analysis, we used the following criteria (Figure 3): (i) the antibodies should exhibit differences in fluorescence intensity at different antibody concentrations. We performed immunofluorescence using two different antibody dilutions (1:500; 1:100); (ii) the subcellular localization of the antibody staining should match that described for the protein; (iii) the antibodies should provide a microscopy image with good

contrast, bright details, and dark background. We find that 14% of mABs failed to work in WB and ICC staining (Supplementary Tables 1 and 3).

6. Any guess of what protein kinase is responsible for phosphorylating T379 of MOR?

Our results regarding phosphorylation of T379 of MOR confirm that this residue is not phosphorylated by PKC. The identity of the kinase responsible for phosphorylation of this site is still not known (reviewed in Lemos Duarte and Devi, 2020).

7. It is unclear how many proteins served as antigens for this study, as much of the description focuses on just a few. Highlighting what proteins in a pathway (cartoon) had mAb generated and validate might be helpful to the reader.

We have added this information to the Figure 1, as well the description of the proteins to Supplementary Table 1.

8. Has the coding sequences of the best mAbs been sequenced? This would ensure renewability.

Yes, all the clones were sequenced prior to being subcloned into full-length IgG1 vectors, then the plasmid preparations re-sequenced to confirm their identity prior to transfection for production from transiently transfected ExpiCHO cells. A master plasmid bank of each heavy and light chain for each of the clones is maintained in -80°C freezers. Prior to each new scale up for a fresh transfection production run, the heavy and light chains are sequenced to confirm identity and the K_D for the cognate peptide antigen is determined by ELISA to confirm that the purified IgG retains the activity observed for the original clone batch. This confers accurate renewability for each clone, which is what differentiates the AvantGen yeast display system from traditional hybridoma technology.

The modifications to the revised manuscript are in blue. I hope that you agree that we have addressed all the concerns of the reviewers and that the article is now acceptable for publication.

Sincerely.

Reference:

- Antal, Corina E., Julia A. Callender, Alexandr P. Kornev, Susan S. Taylor, and Alexandra C. Newton. Intramolecular C2 Domain-Mediated Autoinhibition of Protein Kinase C BII. *Cell Reports* 12:1252–60, 2015
- Lemos Duarte, Mariana, and Lakshmi A. Devi. Post-Translational Modifications of Opioid Receptors. *Trends in Neurosciences* 1–16, 2020
- Pena, Darlene Aparecida, Victor Piana De Andrade, Gabriela Ávila Fernandes Silva, José Ivanildo Neves, Paulo Sergio Lopes De Oliveira, Maria Julia Manso Alves, Lakshmi A. Devi, and Deborah Schechtman. Rational Design and Validation of an Anti-Protein Kinase C Active-State Specific Antibody Based on Conformational Changes *Scientific Reports* 6: 1–11, 2016

REVIEWERS' COMMENTS:

Reviewer #1 (Remarks to the Author):

This manuscript is much improved now

Reviewer #3 (Remarks to the Author):

Thank your for responding to the reviewer's criticisms. The additional figure (cartoon) and experimental details were helpful. Again, you have accomplished and documented an impressive amount of effort. Kudos to the team!

The following sentence in the Abstract - Focusing on understudied synaptic proteins, we generated 137 rabbit monoclonal antibody clones against a panel of antiensies. - has two problems. First, I would call these recombinant antibodies and not monoclonals or monoclonal antibody clones. Perhaps rAb is better for recombinant (or rabbit) antibodies. Monoclonals have a different meaning. Second, I think "antiensies" is meant to be "antigens."

Maybe a regional difference in vocabulary, but usually I usually think about items "fell into," instead of "fell in," to different bins.